# Point-of-Care Wound Blotting with Alcian Blue Grading versus Fluorescence Imaging for Biofilm Detection and Predicting 90-Day Healing Outcomes

**DOI:** 10.3390/biomedicines10051200

**Published:** 2022-05-22

**Authors:** Yu-Feng Wu, Yu-Chen Lin, Hung-Wei Yang, Nai-Chen Cheng, Chao-Min Cheng

**Affiliations:** 1Division of Plastic Surgery, Department of Surgery, National Taiwan University Hospital, Hsin-Chu Branch, Hsinchu 300, Taiwan; fishbee.wu@gmail.com; 2Institute of Biomedical Engineering, National Tsing Hua University, Hsinchu 300, Taiwan; linseal1009@gmail.com; 3Division of Plastic Surgery, Department of Surgery, National Taiwan University Hospital, Biomedical Park Branch, Zhubei City 302, Taiwan; bullskoala@hotmail.com; 4Division of Plastic Surgery, Department of Surgery, National Taiwan University Hospital and College of Medicine, Taipei 100, Taiwan

**Keywords:** chronic wound, biofilm, modified wound blotting with Alcian blue grading, MolecuLight *i:X*, rapid diagnosis, wound healing, point-of-care

## Abstract

Biofilm infection has been identified as a crucial factor of the pathogenesis of chronic wound, but wound biofilm diagnosis remains as an unmet clinical need. We previously proposed a modified wound blotting technique using Alcian blue staining for biofilm detection that was characterized as being non-invasive, time-saving, non-expansive, and informative for biofilm distribution. In this study, we adapted a novel Alcian blue grading method as the severity of biofilm infection for the wound blotting technique and compared its biofilm detection efficacy with MolecuLight *i:X-* a point-of-care florescence imaging device to detect bacteria and biofilm in wounds. Moreover, their predictive value of complete wound healing at 90 days was analyzed. When validated with wound culture results in the 53 enrolled subjects with chronic wounds, the modified wound blotting method showed a strong association with wound culture, while MolecuLight *i:X* only exhibited a weak association. In predicting 90-day wound outcomes, the modified wound blotting method showed a strong association (Kendall’s tau value = 0.563, *p* < 0.001), and the wound culture showed a moderate association (Spearman’s rho = 0.535, *p* < 0.001), but MolecuLight *i:X* exhibited no significant association (*p* = 0.184). In this study, modified wound blotting with the Alcian blue grading method showed superior value to MolecuLight *i:X* both in biofilm detection and predictive validity in 90-day wound-healing outcomes.

## 1. Introduction

Chronic and poor healing wounds are an increasingly problematic and prevalent healthcare issue driven at least in part by a rapidly aging population. Biofilm can be detected in 78% of chronic wounds, which provide an ideal microenvironment for biofilm formation, but only 6% of all acute wounds [1,2]. Biofilms represent a resistant barrier to chronic wound treatment and are highly associated with delayed wound healing. Biofilms are typically composed of 10% bacterial components, such as microcolonies or bacterial clumps, embedded within extracellular polymeric substances (EPSs) that make up the remaining 90%. The EPS not only serves as a physical barrier but also provides communication, known as quorum sensing, between bacteria [3]. Within a mature biofilm, the bacteria remain in a low metabolic state with low mobility, but they exhibit high resistance to the host’s immune defense system and antimicrobial agents. The biofilm state bacteria in the infected wound were estimated to be as much as 500 times more resistant to antibiotics than planktonic bacteria [4].

Although the importance of biofilm infection in delayed wound healing has been well established, biofilm diagnosis in wound patients remains a clinical challenge [5]. In 2017, the Global Wound Biofilm Expert Panel suggested no routinely used diagnostic tools and listed seven clinical indicators for biofilm infected wounds: (1) recalcitrance to treatment with antibiotics or antiseptics; (2) treatment failure despite using appropriate antibiotics or antiseptics; (3) delayed healing; (4) cycles of recurrent infection/exacerbation; (5) excessive moisture and wound exudate; (6) low-level chronic inflammation; and (7) low-level erythema. However, these clinical symptoms/signs can be misleading even for experienced wound specialists. Hence, a reliable, non-biased diagnostic tool for biofilm detection is highly desired.

In 2019, Wu et al. classified current biofilm diagnostic methodology into three primary assay types: morphology, microbiology, and molecular assays [3]. Morphology assays include tissue sampling with histology, scanning electron microscopy (SEM), and confocal laser scanning microscopy (CLSM). However, a representative biofilm histology, including the examination of bacterial clumps and deposition of EPS, may be difficult to perform when using a regular light microscope. The European Society of Clinical Microbiology and Infectious Diseases (ESCIMD) study group for biofilms (ESGB) had recommended SEM and CLSM as the most reliable tools for biofilm diagnosis [6], but these approaches are expensive, time-consuming, and usually not accessible in a clinical setting. Microbiological wound culture is the most widely used and straightforward method to identify possible pathogen presence. Recent studies have shown that wound swabbing using the Levine method is adequate and representative to obtain wound culture, and it is superior to the use of Z technique [7,8]. However, this approach would not detect bacteria in viable but non-culturable (VBNC) states [3,9]. Molecular assays utilize sequences of 16S ribosomal RNA (16S rRNA), which can provide species-specific information for pathogen identification and can detect bacteria in VBNC states. However, these methodologies could not differentiate between planktonic and biofilm bacteria and could be interfered with by genetic material of nonviable bacteria. Furthermore, they do not provide any information regarding the sensitivity of pathogens to antibiotics [3].

Because none of the aforementioned methods can provide point-of-care wound biofilm detection with high diagnostic value or widespread clinical availability, several technologies dedicated to biofilm detection are emerging. These include bacterial florescence imaging devices that can detect wound bacteria load over 10^4^ colony-forming units/gram (CFU/g) and wound blotting methods for biofilm detection [3]. MolecuLight *i:X* is a handheld bacterial fluorescence imaging device that can emit a safe violet light (405 nm) to excite red fluorescence by most bacteria that produce porphyrin, and cyan florescence by *Pseudomonas* species that produce pyoverdine. However, certain bacteria (e.g., *Streptococcus* and *Enterococcus*) do not emit detectable fluorescence [10]. Although this device was initially designed to detect a wound bacteria load over 10^4^ CFU/g, it has also been proposed to detect bacterial biofilm in animal models [11].

Another emerging diagnostic method, wound blotting, has been proposed by Schultz et al. and Nakagami et al. to detect biofilm infection over wounds [12,13,14,15]. This non-invasive approach employs a nitrocellulose membrane to collect wound exudate and Alcian blue or Ruthenium red as a specific biofilm staining agent to target polysaccharides, the primary component of EPS, within biofilms. In our modified wound blotting method using a nylon membrane, good sensitivity of 95.2% could be achieved [16]. The wound blotting protocol can be completed within minutes in most clinical scenarios, such as outpatient clinics, operating rooms, or other healthcare facilities. This approach can provide point-of-care detection of wound biofilm and the information regarding biofilm distribution over the wound bed, which would be beneficial for guiding biofilm-based wound care (BBWC) for subsequent wound management.

In this study, we enrolled chronic wound patients for biofilm detection, using the modified wound blotting method with a novel Alcian blue grading system and fluorescence imaging with MolecuLight *i:X*. Validated by standard wound microbiological culture, the results were compared for biofilm detection efficacy and further analyzed for predicting 90-day wound outcomes.

## 2. Materials and Methods

### 2.1. Study Design

In this prospective cohort study, we performed modified wound blotting in hard-to-heal wounds and classified the results of Alcian blue staining into 4 grades according to the intensity. To compare the efficacy of biofilm detection between our modified wound blotting method and commercialized wound bacterial florescence imaging device (MolecuLight *i:X*), we used initial microbiological wound culture results for validation of biofilm detection. We also compared 3 different methods for predicting 30-day and 90-day wound-healing outcomes: (1) modified wound blotting with Alcian blue grading, (2) microbiological wound culture results, and (3) MolecuLight *i:X*. At first visit, the wound was cleaned and rinsed with normal saline, followed by bacterial florescence imaging, wound blotting, and microbiological culture via wound-swabbing. After conservative debridement, the wound was then dressed according to wound condition and exudate volume. Additional information of wound size at initial debridement and post-debridement 2-week and 4-week was also gathered for outcome evaluation.

### 2.2. Participants

We recruited patients with hard-to-heal wounds that came to National Taiwan University Hospital, Hsin-Chu branch, from September 2020 to September 2021. Patients with wounds that persisted for at least 30 days were enrolled after informed consent. Patients who were less than 20 years of age, pregnant, or not willing to join the trial were excluded. All participants received wound evaluations at their first outpatient clinic visit, and conservative sharp debridement was performed by a plastic surgeon, using surgical scissors, scalpels, and curette to remove necrotic and non-viable tissue. The participants received follow-up examinations every 2 weeks for at least 90 days. The Institutional Review Board of National Taiwan University Hospital, Hsin-Chu Branch (IRB No. 108-105-E), approved our protocol for human specimen gathering and examination.

### 2.3. Wound Evaluation and Outcome Measurement

Wound evaluation included an examination and recording of wound characteristics, including wound size, as well as the results from microbiological wound culturing, modified wound blotting with Alcian blue staining protocol, wound bacterial florescence imaging, and standard wound imaging. We applied transparent film dressing (Tegaderm, 3M, St. Paul, MN, USA) to the wound bed and outlined the wound margin. The photographs of the films were used to estimate the wound size d, using ImageJ software (National Institutes of Health, Bethesda, MD, USA). At 2- and 4-week post-debridement follow-up, percentage of wound size reduction was calculated by the following formula: (baseline wound size - post-debridement 2-week wound size)/baseline wound size × 100 (%). Wound healing is defined as complete epithelization of the wound by inspection. Wound evaluation and wound-healing outcomes were examined and recorded at each visit.

### 2.4. Bacterial Culturing and Identification Methods

Microbiological wound culture was performed by semi-quantitative wound-swabbing via the Levine method, which was superior to the Z technique, and involved rotating the swab over a 1 cm^2^ area of the wound [7,8]. MALDI-TOF rapid diagnostic system (Microflex-LRF MALDI-TOF, Bruker Daltonics GmbH & Co. KG, Bremen, Germany) was utilized for bacterial identification. The MALDI-TOF system involved adding matrix into sample, heated with laser, desorbed, and formed ionized molecules, which then fly into the time-of-flight tube based on their size and charge. The time of flying created a spectrum based on the size and charge of the molecules, which can be matched up with spectral libraries of known organisms [17].

### 2.5. Modified Wound Blotting with Alcian Blue Grading Protocol

Wound blotting was performed before debridement, as illustrated in Figure 1A. After wounds were cleaned and rinsed with non-irritable cleanser and normal saline, sterilized nylon transfer membranes (Biodyne B Nylon Membrane, PALL) were applied and firmly pressed for 10 s over the wound bed for sample loading. In our modified wound blotting procedure, the transfer membrane was soaked in cetyltrimethyl ammonium chloride (CTAC, Emperor chemical, Taipei, Taiwan) for 30 s, with shaking; samples were then and stained with 5 mg/mL Alcian blue 8GX solution (Sigma-Aldrich, St. Louis, MI, USA) for 30 s, followed by soaking in CTAC washing solution for 30 s, with shaking. After air-drying, polysaccharides, the primary biofilm component, were visibly blue. The grading of wound blotting result was determined via direct judgement by two trained observers. Grade 0 represented a negative result, with no visible blue dye on the transfer membrane, while grade 1 to grade 3 represented positive results with increasing blue-color intensity (Figure 1B).

### 2.6. Bacterial Florescence Imaging with MolecuLight i:X

MolecuLight *i:X* (MolecuLight Inc., Toronto, ON, Canada) can emit safe violet light (405 nm) to excite porphyrin-producing bacteria to produce red fluorescence and pyoverdine-producing bacteria (*Pseudomonas* spp.) to produce cyan fluorescence. The surrounding lights were turned off to maintain darkness, and the device was held close to the target wound (approximately 10 cm from the wound) until two indicator lights turned green, implying appropriate conditions for florescence imaging. The presence of red or cyan florescence meant a positive bacterial florescent result, while green, white, or no fluorescence represented negative results.

### 2.7. Statistical Analysis

All statistical analyses were performed by using SPSS v.26.0 software (IBM, Armonk, NY, USA). Descriptive statistics are presented as *n* (%) or mean value (standard deviation). The correlations between modified wound blotting results, microbiological wound culture, bacterial florescence imaging, and wound-healing outcomes were assessed by Pearson’s chi-square test or Spearman’s rank correlation test. The Kruskal–Wallis test was used to evaluate the grading of modified wound blotting with 2-week and 4-week wound-size reduction. To analyze healed and unhealed groups at 90 days, Student’s *t*-test was used for data with normal distribution, and the Mann–Whitney U test was employed for those without normal distribution. The correlation between 4-week decreased wound size and 90-day wound-healing outcomes was assessed by Pearson chi-square test. Statistical significance was considered at *p* < 0.05.

## 3. Results

From September 2020 to September 2021, we enrolled 53 cases with unhealed chronic wounds. The mean age was 65.1 years old, with ages ranging from 21 to 94 years. The primary etiologies of wounds were diabetes (*n* = 19) and trauma (*n* = 19). In Caucasians, venous ulcer was the leading etiology for chronic wounds [18]. However, in Asians, diabetes appeared to be a much more common etiology, followed by pressure injury and arterial insufficiency [19]. Because our hospital was one of the trauma centers in the north of Taiwan, difficult wounds caused by trauma were referred to our clinics, and this may elevate traumatic wounds into our primary etiology. Regarding wound examination results before initial debridement, 40 cases (75.5%) yielded positive wound culture results, 44 cases (83%) yielded positive wound blotting results, and 19 cases (35.8%) showed positive bacterial florescence. The most common pathogen was *Staphylococcus aureus* (34.0%), followed by *Enterococcus* species and *Pseudomonas* species. Most wounds showed improvement, except for gradually deteriorating and enlarging wounds in four participants during the follow-up period. One of the four participants required below-knee amputation 33 days after the first debridement, due to poor wound condition. Detailed characteristics of participants are described in Table 1. Representative positive and negative cases are illustrated in Figure 2.

### 3.1. Biofilm Detection by Modified Wound Blotting with Alcian Blue Grading versus MolecuLight i:X

Our modified wound blotting evaluation before debridement revealed 9 cases classified as grade zero (negative result), 7 cases as grade one, 16 cases as grade two, and 21 cases as grade three, respectively. When validated with microbiological wound culture data, grade-two and grade-three cases had 93.8% and 95.2% positive results, and grade-zero cases had 100% negative results, which demonstrated a significant and strong association between wound blotting grading and microbiological culture results (Spearman’s rho = 0.641, *p* < 0.001; Table 2).

Regarding MolecuLight *i:X* examination before debridement, all positive cases also had positive modified wound blotting results (grades 1–3). When validated with initial microbiological wound culture, 94.7% of cases yielded positive wound cultures in the positive bacterial fluorescence group (Table 2). However, in the negative bacterial fluorescence group, 64.7% of cases had positive wound culture, and only 35.3% had negative results. This finding suggested low sensitivity (45%) but high specificity (92.3%) and high positive predictive value (94.7%) for MolecuLight *i:X* to indicate wound infection.

### 3.2. Predicting 90-Day Wound-Healing Outcomes by Modified Wound Blotting with Alcian Blue Grading Versus MolecuLight i:X

Regarding long-term wound-healing outcomes at the 90-day follow-up, no participants with an initial grade-two or grade-three staining had healed wounds before 30 days, 42.9–50% of them had healed wounds between 30 and 90 days after initial debridement, and 50–57.1% of wounds remained unhealed at the 90-day follow-up. Among those with an initial grade-zero or grade-one staining by wound blotting, all wounds healed within 90 days. In grade-one cases, 14.3% of cases healed within 30 days, and 85.7% healed between 30 and 90 days. In grade-zero cases, most cases (77.8%) healed within 30 days, and 22.2% healed between 30 and 90 days. Statistically, this demonstrates a strong association between the grading of wound blotting and 90-day wound-healing outcomes (Kendall’s tau value = 0.563, *p* <0.001; Table 3). When using wound microbiological culture results for predicting wound outcomes, we noted that 50% of all cases in the positive wound culture group healed within 30–90 days, and 47.5% remained unhealed after 90 days. In the negative wound culture group, most cases (53.8%) healed within 30 days, 38.5% healed during the 30–90-day period, and only 7.7% cases remained unhealed at 90 days. A moderate association between initial wound culture results and 90-day wound outcomes was noted (Spearman’s rho = 0.535, *p* < 0.001; Table 3).

When using bacterial fluorescence images to predict 90-day wound outcomes (Table 3), positive bacterial fluorescence results were associated with healed wounds during the 30–90-day period (57.9% of all cases). The remaining 42.1% of all cases remained unhealed at 90 days. However, no significantly better wound-healing outcomes were associated with the negative bacterial fluorescence result group (*p* = 0.184).

Although our modified wound blotting method for biofilm detection had showed promising results in biofilm detection and the prediction of wound-healing outcome, there were still other predisposing factors related to prolonged wound healing. In Figure 3, we illustrate an example of a biofilm-negative case with relatively prolonged wound healing due to tunneling and dead space under the wound.

### 3.3. Correlation between Modified Wound Blotting with Alcian Blue Grading and Post-Debridement Wound Decreased Size

To assess the correlation between changes in wound size and modified wound blotting with Alcian blue staining, we recorded wound size before debridement and at 2 weeks and at 4 weeks following initial debridement (Table 4). We found that cases with an initial grade 0 staining by wound blotting demonstrated 79.4% and 94.5% wound-size reduction at the post-debridement 2 weeks (POW2) and post-debridement 4 weeks (POW4), respectively. However, grade-two and grade-three cases demonstrated only 34–36.6% and 48.1–53% wound-size reduction at POW2 and POW4, respectively. This indicates a significant difference when comparing grade-zero data with grade-two or -three data at POW2 (grade two, *p* = 0.018; grade three, *p* = 0.013) and POW4 (grade two, *p* = 0.001; grade three, *p* = 0.002).

### 3.4. Risk Factor Analysis in 90-Day Wound-Healing Outcomes

When comparing the 90-day healed group with 90-day unhealed group, a significant higher prevalence of diabetes, lower POW2 and POW4 wound decreased size, higher positive wound cultures, higher positive result of modified wound blotting and Alcian blue staining, and higher positive results of MolecuLight *i:X* in the 90-day unhealed group were observed (*p* < 0.001) (Table 5). The average wound-size decrease ratio at POW4 was 86.14% in the healed group, but only 22.81% was noted in the unhealed group. By using the wound-size decrease ratio greater than 50% at POW4 to predict the 90-day wound-healing outcomes, high sensitivity (96.96%) and a high negative predictive value (94.1%) were noted. Moreover, 90-day wound-healing outcomes showed significant differences between groups for the POW4 wound-size decrease ratio greater and less than 50% (Table 5).

## 4. Discussion

This is the first study aimed at comparing the wound blotting method and MolecuLight *i:X* in biofilm detection and the first time that adapted Alcian blue grading system in the wound blotting method to evaluate the severity of biofilm infection. The wound blotting method for biofilm detection is characterized as a non-invasive, cheap, and time-saving technique with high predictive value. On the other hand, MolecuLight *i:X* is a real-time fluorescence imaging device that was initially designed to detect moderate-to-heavy growth of bacteria with bacterial loads greater than 10^4^ CFU/g and has been proposed to assist in the detection and removal of wound biofilm in animal models [11,20]. Although our wound blotting method is not truly real time as MolecuLight *i:X* is, it takes only a few minutes to get the report. Moreover, when using MolecuLight *i:X*, we need to keep the surroundings in darkness and keep the device quite close to the target wound, and this may increase the risk of contamination during the procedures. Additionally, the detection mechanism of MolecuLight *i:X* is still based on the detection of bacterial florescence within biofilm rather than biofilm itself, so this may contribute to false-negative results when the biofilm presents with insufficient bacterial load or when the biofilm formed by bacteria without detectable florescence. Considering that EPS accounts for 90% of biofilm components and bacteria account for only 10%, we directly targeted the polysaccharides, the most abundant component of EPS, in our modified wound blotting method for biofilm detection.

Both modified wound blotting with Alcian blue grading and MolecuLight *i:X* can provide additional information regarding biofilm or bacterial distribution over the wound bed and peri-wound region that can be used for targeting wound debridement. Several studies have demonstrated that wound debridement and management with the aid of bacterial florescence can accelerate the wound-healing rate (23% increase within 12-weeks), reduce antibiotic prescription needs, decrease antimicrobial dressing use, and save 10% of annual wound costs [10,21]. In our experience, when comparing the modified wound blotting method with MolecuLight *i:X*, we found that the former had higher sensitivity in biofilm or bacterial distribution, as can be seen in our representative case with positive results (Figure 2A).

In the clinical use of MolecuLight *i:X,* one deficit is that skin and tissue will emit green fluorescence, which makes it difficult to distinguish from the cyan fluorescence associated with *Pseudomonas* infection. In a burn-wound study that used fluorescence imaging, cyan fluorescence resulted in a sensitivity of 100% for the detection of *Pseudomonas*, but only 44% positive predictive value (PPV), which may be related to difficulties interpreting the green florescence of wound tissue compared to the cyan [22]. Several criteria have been proposed to guide fluorescence signal interpretation: (1) the fluorescent signature should have a glowing white center with a blue/green border, (2) the cyan fluorescence observed should not correspond to any specific landmark or tissue structure on a standard image, and (3) the color of the surrounding skin will appear dull green rather than bright white/cyan [23]. Although several in vitro and preclinical animal studies had proposed the utility of MolecuLight *i:X* in biofilm detection, it remains unclear whether bacteria embedded within biofilm that remain in low metabolic status will interfere with the detectable bacterial load or alter the results of fluorescence staining and imaging approaches to detect and treat wounds [11,24]. In our study, MolecuLight *i:X* demonstrated high PPV (94.7%) but low sensitivity (45%) when validated against microbiological wound culture results. This may be due to cyan florescence being hindered by wound tissue that emits green florescence, or insufficient bacterial load in chronic wounds to support fluorescence imaging. Among 19 positive cases, six cases demonstrated cyan florescence, including three cases that demonstrated red and cyan fluorescence simultaneously. Among cases with positive wound cultures, four cases yielded *Pseudomonas* species and all of them demonstrated positive cyan fluorescence.

In this study, we observed different staining intensities within positive wound blotting results. We were curious about the correlation between the staining intensity, the severity of biofilm infection, and the wound-healing outcome. We therefore classified wound blotting results into four grades. Grade zero represents negative staining results, and positive results from grade one to grade three are ranked according to staining intensity, as determined by direct visual observation. To facilitate the wound blotting examination process and provide rapid results, stain grading by direct visual observation, versus the use of Image J software, is preferable in terms of time and cost. While we took into account the fact that different bacteria may be related to different levels of polysaccharide production, most of our positive wound cultures were polymicrobial (62.5%) rather than monomicrobial (37.5%). Therefore, we used only positive or negative culture results when analyzing their correlation with staining intensity. Table 2 illustrates the signification correlation between wound blot grading and culture results (*p* < 0.001). We found that grade-two and -three staining results were associated with high positive wound culture rates of 93.8% and 95.2%, but a few cases demonstrated negative culture results. This may be due to viable but non-culturable (VBNC) bacteria in biofilm, or bias related to swabbing methods. It may also be related to the fact that bacteria had characteristically low motility, high attachment, and were deeply embedded within the biofilm structures of mature biofilms. Among all wound culture methods, culture with tissue biopsy has been proven to be the most reliable but invasive approach. However, wound swabbing via the Levine technique or the Z technique has been proposed as an alternative and non-invasive approach with good results. Comparing the two swabbing methods, we see that recent trials have established that the Levine technique is superior to the Z technique [7,8].

When comparing the three methods, namely modified wound blotting with four grades, microbiological wound culture, and MolecuLight *i:X*, for predicting 90-day wound-healing outcomes, modified wound blotting with Alcian blue grading system demonstrated a strong association, and microbiological wound culture demonstrated a moderate association. This result indicated that the abundance of blotted biofilm component, described as grading of Alcian blue intensity, can represent the severity of biofilm infection, which related to the 90-day wound-healing outcomes. In the MolecuLight *i:X* study, we found no significant correlation for the 90-day healing outcome (*p* = 0.184). This may be attributed to low sensitivity and low negative predictive value (NPV = 35.3%). However, a similar distribution of healing periods for grades two and three Alcian blue wound blotting results could be observed in the positive fluorescence group.

Regarding the relationship between the grading of wound blotting and wound-size reduction at POW2 and POW4, we noticed that higher grading of wound blotting results seems to be related to a smaller wound-size decrease at POW2 and POW4. However, a statistical significance can be seen only when comparing grade zero to grade two and grade three, and this may be the result of the limited case number.

Sheehan et al. published a large multicenter prospective randomized controlled trial to illustrate that the reduction of wound size by 4 weeks can well predict the healing outcomes by 12 weeks in diabetic ulcers [25]. The cutoff value of wound reduction by the 4th week was 53%, those exceeding this value had 58% cases healed by 12 weeks, but only 9% of cases healed in those below-cutoff values. In subsequent studies, many authors had conducted similar results and confirmed 50% wound reduction by 4 weeks as a strong predictor of wound healing by 3 months [26,27,28]. Our cases, which included all types of chronic wounds, also showed similar results that a POW4 wound reduction greater than 50% was a significant predictor of 90-day wound-healing outcomes (Table 5B).

There are some limitations to our wound blotting and biofilm detection method. This method could not identify the pathogen, a deficit that might be compensated for by microbiological wound culture, and it demonstrated a relatively lower detection rate when applied to dry gangrene or dry eschar, which may be related to sample loading difficulties. Additionally, biofilm infection is an essential but not the only one risk factor of chronic wounds, as illustrated in the representative case with negative results but delayed healing (Figure 3). Therefore, further study to eliminate possible bias should be conducted.

## 5. Conclusions

Our modified wound blotting method for biofilm detection with a novel Alcian blue grading system demonstrated a significant and strong correlation to microbiological wound culture results. The Alcian blue grading system directly represented the abundance of biofilm component, which can be interpreted as the severity of biofilm infection, also demonstrated a significant and strong correlation to 90-day wound-healing outcomes. This point-of-care wound blotting method can not only detect the presence of biofilm, but also the distribution and abundance of biofilm in only a few minutes, which can facilitate biofilm-based wound-care methodology. Additionally, the detection threshold for the wound blotting method was lower than that for bacterial florescence imaging with MolecuLight *i:X*, which basically detected the bacterial component within biofilm. Additional research aimed at validating this novel biofilm detection approach, in comparison to histology or morphology results, should be conducted to improve and confirm its predictive value.

## Figures and Tables

**Figure 1 biomedicines-10-01200-f001:**
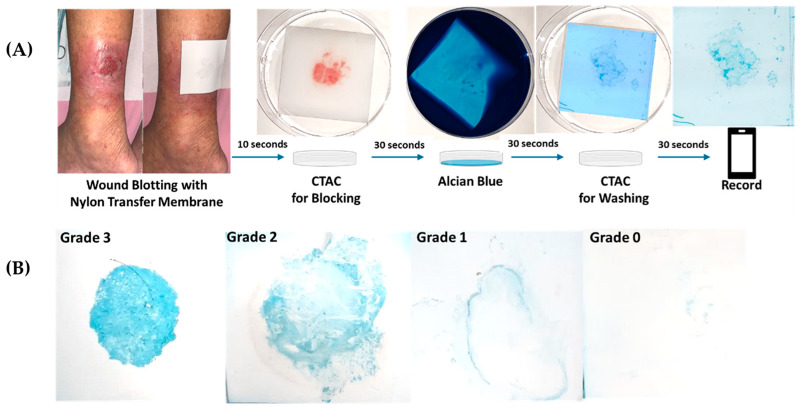
Modified wound blotting with Alcian blue grading algorism: (**A**) modified wound blotting with Alcian blue grading algorism (CTAC: cetyltrimethyl ammonium chloride) and (**B**) representative images of Alcian blue grading.

**Figure 2 biomedicines-10-01200-f002:**
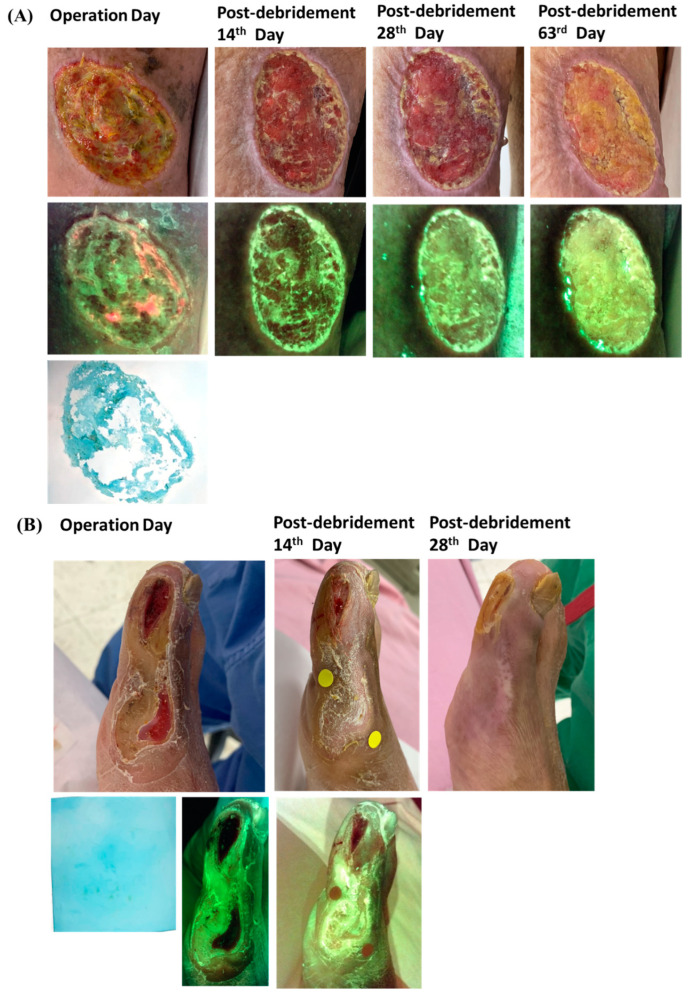
Representative clinical cases: (**A**) Representative positive case: An 82-year-old female had traumatic leg ulcer with polymicrobial wound culture result, modified wound blotting grade 3, and positive MolecuLight *i:X* result. Her wound remained unhealed at 90-day follow-up. (**B**) Representative negative case: A 64-year-old male with foot ulcer related to diabetes and arterial insufficiency. He had negative wound culture, modified wound blotting grade 0, and negative MolecuLight *i:X*. The wound completely healed post-debridement by the 28th day.

**Figure 3 biomedicines-10-01200-f003:**
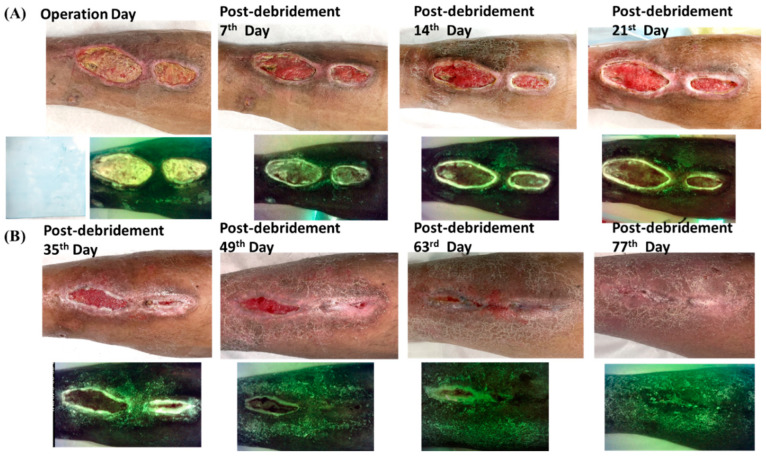
Clinical biofilm-negative case with delayed wound healing: a 44-year-old male with traumatic leg wound. (**A**) Despite a negative wound culture and modified wound blotting and MolecuLight *i:X* results, his wound had poor improvement at the first 1 month due to tunneling and dead space under the wound; (**B**) since the 2nd month, the wound improved promptly after the dead space and tunneling disappeared. His wound finally healed by the 77th day.

**Table 1 biomedicines-10-01200-t001:** Demographic data of chronic wound cases: descriptive statistics are presented as *n* (%) or mean value (standard deviation).

Variables	
**Age (years)**	65.1 (18.4)
**Gender (Male:Female)**	24:29
**Diabetes**	19 (35.9)
**Etiology of wound**	
Diabetic	19 (35.9)
Arterial insufficiency	11 (20.8)
Venous ulcer	4 (7.5)
Pressure injury	5 (9.4)
Trauma	19 (35.9)
Other	2 (3.8)
**Location of wound**	
Forefoot	17 (32.1)
Midfoot	3 (5.7)
Hindfoot	7 (13.2)
Leg	19 (35.9)
Hip	4 (7.5)
Perineum	1 (1.9)
Hand	2 (3.8)
**Wound Size**	
Initial wound size (cm^2^)	9.51 (14.40)
2-Week Wound Size (cm^2^)	5.13 (7.91)
4-Week Wound Size (cm^2^)	3.46 (7.06)
2-Week Decreased Size (%)	33.96 (31.31)
4-Week Decreased Size (%)	63.00 (44.72)
**Wound Culture (+)**	40 (75.5)
Mono-microbial	15 (37.5)
Poly-microbial	25 (62.5)
2 species	12 (30.0)
≥3 species	13 (32.5)
**Grade of Wound Blotting**	
Grade 0	9 (17.0)
Grade 1	7 (13.2)
Grade 2	16 (30.2)
Grade 3	21 (39.6)
**MolecuLight *i:X* (+)**	19 (35.8)
**Wound-Healing Time**	
<30 Days	8 (15.1)
30–90 Days	25 (47.2)
>90 Days	20 (37.7)

**Table 2 biomedicines-10-01200-t002:** Modified wound blotting with Alcian blue grading and MolecuLight *i:X* in predicting wound culture results: Modified wound blotting had strong association with wound culture results (Spearman’s rho = 0.641, *p* < 0.001); MolecuLight *i:X* only showed weak association with wound culture results (Phi (φ) = 0.335, *p* = 0.015), and poor sensitivity in biofilm detection.

Alcian Blue Grading	Wound Culture (+)	Wound Culture (−)	Case No.
**Grade 3**	20 (95.2)	1 (4.8)	21
**Grade 2**	15 (93.8)	1 (6.3)	16
**Grade 1**	5 (71.4)	2 (28.6)	7
**Grade 0**	0 (0)	9 (100)	9
	**Wound Culture (+)**	**Wound Culture (−)**	**Case No.**
**M** **olecuLight *i:X* (+)**	18 (94.7)	1 (5.3)	19
**M** **olecuLight *i:X* (** **−** **)**	22 (64.7)	12 (35.3)	34
**M** **olecuLight *i:X* in Biofilm Detection**			
Sensitivity			45%
Specificity			92.3%
Positive Predictive Value			94.7%
Negative Predictive Value			35.3%

**Table 3 biomedicines-10-01200-t003:** Three methodologies in predicting 90-day wound-healing outcomes: modified wound blotting with Alcian blue grading had strong association with 90-day wound outcomes (Kendall’s tau value = 0.563, *p* < 0.001); wound culture results had moderate association with 90-day wound outcomes (Spearman’s rho = 0.535, *p* < 0.001); and MolecuLight *i:X* showed no significant association with 90-day wound outcomes (Spearman’s rho = 0.185, *p* = 0.184).

**Alcian Blue Grading**	**<30 Days Healing**	**30–90 Days Healing**	**>90 Days Healing**	**Case No.**
**Grade 3**	0 (0)	9 (42.9)	12 (57.1)	21
**Grade 2**	0 (0)	8 (50.0)	8 (50.0)	16
**Grade 1**	1 (14.3)	6 (85.7)	0 (0)	7
**Grade 0**	7 (77.8)	2 (22.2)	0 (0)	9
	**<30 Days Healing**	**30–90 Days Healing**	**>90 Days Healing**	**Case No.**
**Wound Culture (+)**	1 (2.5)	20 (50.0)	19 (47.5)	40
**Wound Culture (−)**	7 (53.8)	5 (38.5)	1 (7.7)	13
	**<30 Days Healing**	**30–90 Days Healing**	**>90 Days Healing**	**Case No.**
**MolecuLight *i:X* (+)**	0 (0)	11 (57.9)	8 (42.1)	19
**MolecuLight *i:X* (−)**	8 (23.5)	14 (41.2)	12 (35.3)	34

**Table 4 biomedicines-10-01200-t004:** Correlation between modified wound blotting results and post-debridement 2-week and 4-week wound decreased size. Statistically significant differences could be seen both in grade 3 vs. grade 0 and grade 2 vs. grade 0 at post-debridement 2-week and 4-week time points.

**Alcian Blue Grading**	**Post-Debridement 2-Week Decreased Size (%)**	**Post-Debridement 4-Week Decreased Size (%)**	**Case No.**
**Grade 3**	34.0	48.1	21
**Grade 2**	36.6	53.0	16
**Grade 1**	55.3	86.9	7
**Grade 0**	79.4	94.5	9
**Alcian Blue Grading**	**Post-Debridement 2-Week Decreased Size (%)**	**Post-Debridement 4-Week Decreased Size (%)**	
Grade 3 vs. Grade 0	*p* = 0.013	*p* = 0.002	
Grade 2 vs. Grade 0	*p* = 0.018	*p* = 0.001	

**Table 5 biomedicines-10-01200-t005:** Comparing between 90-day healed and 90-day unhealed groups for possible risk factor evaluation. Descriptive statistics are presented as *n* (%) or mean value (standard deviation). Besides, post-debridement 4-week wound reduction over 50% was recognized as a good predictor of 90-day wound healing outcomes (*p* < 0.001) with good sensitivity and specificity.

Variables	90-Day Healed	90-Day Unhealed	*p*-Value
**Age (year)**	61.9 (19.4)	70.5 (15.6)	0.102
**Gender (Male:Female)**	15:18	10:10	
**Diabetes (%)**	27.3	50.0	<0.001
**Wound Size**			
Initial Wound Size (cm^2^)	9.56 (16.11)	9.41 (11.73)	0.970
2-Week Decreased Size (cm^2^)	5.96 (2.39)	1.62 (1.50)	<0.001
2-Week Decreased Size (%)	63.14 (25.41)	18.41 (17.05)	<0.001
4-Week Decreased Size (cm^2^)	8.05 (1.54)	2.56 (5.61)	<0.001
4-Week Decreased Size (%)	86.14 (16.43)	22.81 (50.01)	<0.001
**Wound Culture (+)**	21 (63.6%)	19 (95%)	<0.001
**Alcian Blue Grading (+)**	24 (72.7%)	20 (100%)	<0.001
Grade 0	9	0	
Grade 1	7	0	
Grade 2	8	8	
Grade 3	9	12	
**MolecuLight *i:X* (+)**	11 (33.3%)	8 (40%)	<0.001
**Healing Duration (day)**	45.1 (23.1)	>90	
**Total No.**	33	20	
	**90-Day Healed**	**90-Day Unhealed**	**Case No.**
**4-Week Decreased Size > 50%**	32 (88.9%)	4 (11.1%)	36
**4-Week Decreased Size < 50%**	1 (5.9%)	16 (94.1%)	17
**“4-Week Decreased Size > 50%” in predicting 90-Day Healing**			
Sensitivity			96.96%
Specificity			80.0%
Positive Predictive Value			88.9%
Negative Predictive Value			94.1%

## Data Availability

The datasets generated during and/or analyzed during the current study are available from the corresponding author upon reasonable request.

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
