# Peer review of "Point-of-Care Wound Blotting with Alcian Blue Grading versus Fluorescence Imaging for Biofilm Detection and Predicting 90-Day Healing Outcomes"

_biomedicines, 2022, doi:10.3390/biomedicines10051200_

Round 1
Reviewer 1 Report
The present manuscript describes a modified wound blotting technique using an Alcian blue stain that can detect biofilms in wounds. This technique is compared to MolecuLight (a comparable florescence imaging device). Non-healing wounds are assessed over a 90 day period.
Major comments:
Bacterial culturing and identification methods are missing in the methods section. please add
Mild formatting issue on tables. On some tables the table number and title are on top of the table and other times below. Make consistent (top of table)
Figure 3 neds to be moved into results and discussed there first, instead of presenting new data in the discussion. It can still be discussed in the discussion as a limitation but no new data should be presented in the discussion section.
Minor comments:
Line 20: compare should be compared
Table 1: 'others' should be other
Author Response
Major comments:
Point 1:.Bacterial culturing and identification methods are missing in the methods section. please add
Response 1: Thanks for your suggestion, we have added a paragraph to demonstrate our bacterial culturing and identification methods in the Materials and Methods section.
Point 2: Mild formatting issue on tables. On some tables the table number and title are on top of the table and other times below. Make consistent (top of table)
Response 2: Thanks for your suggestion, we have revised our formatting of tables to make them consistent.
Point 3: Figure 3 needs to be moved into results and discussed there first, instead of presenting new data in the discussion. It can still be discussed in the discussion as a limitation but no new data should be presented in the discussion section.
Response 3: Thanks for your comment, we have moved figure 3 to the “Results” section and described there.
Minor comments:
Point 1:. Line 20: compare should be compared
Response 1: we have revised the grammatical error.
Point 2: Table 1: 'others' should be other
Response 2: we have revised the grammatical error.

Reviewer 2 Report
Overall, I found this manuscript highly interesting with regards to the detection of biofilms in chronic wounds using Alcian blue. From a general standpoint, the manuscript contains a few grammatical errors throughout the article, hence, I suggest a thorough proofread prior to resubmission.
My general comments are as follows:
- line 196 - the word 'species' should not be italicised. Please amend.
- line 286 - first sentence is grammatically incorrect. Please amend.
- line 344 - 'in vitro' is Latin, and hence by convention, should be italicised.
- line 354 - Pseudomonas should be italicised.
- paragraph commencing on line 413 - I would suggest deleting it as it does not value add to the manuscript, but instead, detracts.
Author Response
Point 1:. Overall, I found this manuscript highly interesting with regards to the detection of biofilms in chronic wounds using Alcian blue. From a general standpoint, the manuscript contains a few grammatical errors throughout the article, hence, I suggest a thorough proofread prior to resubmission.
Response 1: Thanks for your comments, we have reviewed and revised the grammatical errors of our manuscript.
General comments:
Point 1:. Line 196 - the word 'species' should not be italicised. Please amend.
Response 1: Thanks for the comment, we have revised the format.
Point 2: line 286 - first sentence is grammatically incorrect. Please amend.
Response 2: Thanks for the comment, we have revised the sentence for grammatical error.
Point 3: line 344 - 'in vitro' is Latin, and hence by convention, should be italicised.
Response 3: Thanks for the comment, we have revised into italicized format.
Point 4: line 354 - Pseudomonas should be italicised.
Response 4: Thanks for the comment, we have revised into italicized format.
Point 5: paragraph commencing on line 413 - I would suggest deleting it as it does not value add to the manuscript, but instead, detracts.
Response 5: Thanks for your suggestion, we have deleted this paragraph.

Reviewer 3 Report
The article by Wu and col. proposes a new method to evaluate the severity of biofilm infection in the healing process of chronic wounds. The article is very interesting, it can be seen that the research team has experience in the field; the results are encouraging and certainly show that research can continue in this field.
An observation regarding the wording, in the sense of using the space before [ ( the reference number)
I consider that the article meets the criteria to be published.
Author Response
Point 1:. The article by Wu and col. proposes a new method to evaluate the severity of biofilm infection in the healing process of chronic wounds. The article is very interesting, it can be seen that the research team has experience in the field; the results are encouraging and certainly show that research can continue in this field.
An observation regarding the wording, in the sense of using the space before [ ( the reference number)
I consider that the article meets the criteria to be published.
Response 1: Thanks for your precious comments and suggestions, we have revised our manuscript and added space before the cited reference number as your suggestion.
